# The ADAR Family in Amphioxus: RNA Editing and Conserved Orthologous Site Predictions

**DOI:** 10.3390/genes11121440

**Published:** 2020-11-30

**Authors:** Michał Zawisza-Álvarez, Claudia Pérez-Calles, Giacomo Gattoni, Jordi Garcia-Fernàndez, Èlia Benito-Gutiérrez, Carlos Herrera-Úbeda

**Affiliations:** 1Department of Genetics, Microbiology and Statistics, Faculty of Biology, and Institut de Biomedicina (IBUB), University of Barcelona, 08007 Barcelona, Spain; mihizawi@gmail.com (M.Z.-Á.); claudia05pc@gmail.com (C.P.-C.); jordigarcia@ub.edu (J.G.-F.); 2Department of Zoology, University of Cambridge, Cambridge CB2 1TN, UK; gg456@cam.ac.uk

**Keywords:** RNA-editing, invertebrates, evolution

## Abstract

RNA editing is a relatively unexplored process in which transcribed RNA is modified at specific nucleotides before translation, adding another level of regulation of gene expression. Cephalopods use it extensively to increase the regulatory complexity of their nervous systems, and mammals use it too, but less prominently. Nevertheless, little is known about the specifics of RNA editing in most of the other clades and the relevance of RNA editing from an evolutionary perspective remains unknown. Here we analyze a key element of the editing machinery, the ADAR (adenosine deaminase acting on RNA) gene family, in an animal with a key phylogenetic position at the root of chordates: the cephalochordate amphioxus. We show, that as in cephalopods, ADAR genes in amphioxus are predominantly expressed in the nervous system; we identify a number of RNA editing events in amphioxus; and we provide a newly developed method to identify RNA editing events in highly polymorphic genomes using orthology as a guide. Overall, our work lays the foundations for future comparative analysis of RNA-editing events across the metazoan tree.

## 1. Introduction

Changes in the regulation of gene functions are at the basis of morphological evolution, from the simplest metazoans to the possessors of amazingly complex nervous systems. Contemporary textbook examples of mechanisms that allow exquisite fine-tuning of this regulation are alternative splicing in vertebrates and RNA gene editing in cephalopods [1,2,3]. Both enable an increase of proteome and interactome complexity [4] without affecting genome complexity by addition of extra genes. In this regard, RNA-editing is a post-transcriptional modification that has not received much attention. It adds an extra layer of regulation at the translational level, as well as an increase in proteome diversity without changes in the original DNA sequence. In an RNA-editing event, a specific nucleotide of the RNA transcript undergoes a chemical change. This process transforms the edited nucleotide into a different one, changing the sequence of the messenger RNA molecule and thus, potentially modifying the amino acid sequence [5]. RNA-editing is present in all eukaryotic organisms [6,7,8] in different forms, such as the C-to-U (cytidine to uracil) editing mediated by the Apobec protein family [7] or the most prevalent in metazoans, the A-to-I (adenosine to inosine) editing, mediated by the ADAR (adenosine deaminase acting on RNA) protein family [9].

Although increasing the levels of regulation and the diversity of the proteome has clear benefits, the evolutionary implications of RNA-editing are yet to be discovered. The role of the ADAR family itself is well known in vertebrates and cephalopods [10]. Its removal, for example, is known to be lethal in *Mus musculus*. Mouse embryos lacking ADAR will die before birth due to stress-induced apoptosis [11] and mice with null ADARB1 will become prone to seizures and die young [12]. In humans, one of its roles is the regulation of the innate immunity via the modulation of the antiviral response. In this case, ADAR would edit the viral dsRNA, which in some cases inactivates the virus. However, this also generates mismatched pairs on the viral dsRNA that dampen the interferon response, making ADAR in some cases a pro-viral agent [13]. Several studies have demonstrated how some level of conservation exists on the edited amino acid in vertebrates [14,15,16]; however, since the mechanisms or the features that restrict editing to specific nucleotides remain unknown, we are not yet able to properly predict RNA-editing sites *in silico*. This is partially due to the difficulty of detecting *bona fide* RNA-editing events in enough species, which ideally requires the sequencing of both DNA and RNA from the same individual [17]. This is particularly challenging when dealing with highly polymorphic invertebrate species such as the cephalochordate amphioxus [18]. In this key evolutionary model for the invertebrate/vertebrate transition, and in many other less-established laboratory species, extracting both genomic DNA and RNA in sufficient quantities from the same individual may be difficult to achieve and the sequencing quality required for in-depth RNA-editing studies unaffordable. On the other hand, the study of conservation and changes in RNA-editing in key evolutionary models is crucial to understand if this regulatory mechanism might have had a role in generating morphological novelty.

Here, we present for the first time the characterization of the ADAR family and its expression through the development of the amphioxus *Branchiostoma lanceolatum*, and propose a method to deal with its highly polymorphic genome when identifying in silico RNA-editing sites, that can be used with any organism with genomic and RNA-seq data available. This novel method, combined with an original pipeline, allows the identification of orthologous editing events between two species. Overall, our work lays the foundations for future comparative analysis of RNA-editing events across the metazoan tree.

## 2. Materials and Methods

### 2.1. Phylogenetic Tree of ADAR Family Genes

We selected the following sequences from NCBI: NP_001180424.1, NP_001344887, ALX72798.1, XP_031746878, XP_026690338.1, NP_571671.2, XP_030849200.1, NP_001103.1, NP_001020008, NP_001104544.1, NP_001096190.1, XP_021334694.1, AAF78094.1, NP_443209.2, XP_418563.4, XP_031759801.1, XP_686426.5, XP_781832.1, XP_002128212.1, XP_015888627.1, XP_020412580.1, XP_019848877.1, XP_003387867.1 for alignment with MAFFT (v7.307) [19] alongside the two translated genes from amphioxus BL01761 and BL03778. The aligned sequence was trimmed to select the conserved domain and a maximum likelihood tree was generated using IQ-TREE [20]. The tree is rooted on two plant ADAR-like proteins from *Ziziphus jujuba* and *Prunus persica.*

### 2.2. Third Generation In Situ Hybridization Chain Reaction (HCR) for Detecting Gene Expression

We followed the third generation in situ Hybridization Chain Reaction protocol, which has been recently optimized for amphioxus [21]. We used the following sequences to generate the probes: BL03778 (*Adar*), BL01761 (*Adarb*), BL13719 (*Prox1*), and BL11117 (*Elav*) from the transcriptome produced by Marletaz et al. [18].

### 2.3. Basic Single Nucleotide Variants Filter for RNA-Editing Candidates

In the first step of our pipeline, we used bowtie2 (v2.3.5.1) [22] to align, for each available tissue and developmental stage, the amphioxus RNA-seq reads (accessions SRX7351856 to SRX7351867) to the reference *B. lanceolatum* genome [18] and then the SAMtools library [23] to perform the Single Nucleotide Variants (SNV) analysis. We filtered the SNVs to get only the changes from A to G if the transcript was on the forward strand and T to C if the transcript was on the reverse strand, adding a depth cut-off of 10 and only considering the positions within coding regions. The resulting list of SNVs is the basic candidate list of RNA-editing events.

### 2.4. Using Genomic Reads to Filter out Confirmed SNPs

We repeated the previous steps with the DNA reads (accession PRJEB13665) instead of the RNA-seq reads in order to get the confirmed SNPs. In this step we did not use the depth cut-off. We then filtered out the candidate list of SNVs that appeared as confirmed SNPs in order to get the putative RNA-editing events. We then made an aggregative analysis of all the tissue and stage putative events.

### 2.5. GO Analysis

The analyzed sequences were blasted against the RefSeq protein database and run through InterProScan. Then, the blasted sequences were mapped and annotated using Blast2GO [24] and merged with the InterProScan GO annotations.

### 2.6. Orthologous RNA-Editing with Human Editing

From the candidate list and a human RNA-editing database (REDIportal [25]) we used a list of homologous genes obtained via BLAST to select those candidates that were within genes with human homologues, as well as the human editing targets present in those homologues from the REDIportal database [25]. We then obtained the mRNA sequence of all the isoforms of genes that contained the candidate SNVs that had potential homology with humans and translated the sequences into the amino acid sequence. We also obtained all the amino acid sequences of the homologous human genes from Ensembl [26]. We then used MAFFT (v7.307) [19] to make the multiple alignments of all the homologous amino acid sequences (all amphioxus isoforms with each homologous human isoform). We then checked within the alignments the edited amino acids and selected those cases where the edited amino acids in human were within the accepted window size in the alignment of the candidate SNVs in the amphioxus and separated the cases in categories according to the conservation of the original and edited amino acids. We used accepted window sizes of 0, 1, 5, and infinite amino acids. Finally, we performed the aggregative analysis across window sizes, tissues and stages (Appendix A).

For the curation of the events in window size 0, class 1, we manually evaluated the quality of the alignment within the local region to the putative events. We also used the genomic sequence of the annotated genes to get the full pre-mRNA sequence of all the isoforms that include the putative events. We then got the predicted secondary structure of the pre-mRNA molecules using linearfold [27]. We considered a positive dsRNA result if the position of the putative editing was either in a dsRNA or in a nick (a ssRNA fragment of one nucleotide in length with dsRNA on both sides).

## 3. Results

### 3.1. The European Amphioxus has Two ADAR Genes

Using the available transcriptomic data from *B. lanceolatum* [18], we were able to retrieve two amphioxus genes belonging to the ADAR family. Analyses of these two genes by sequence alignment to homologous genes found in publicly available databases, allowed us to identify these sequences as *Adar* and *Adarb* (Figure 1A). As expected from its pre-duplicative condition, the amphioxus ADAR family was composed of only two members and lacks one of the vertebrate-specific paralogs.

We further exploited available RNA-seq datasets to characterize the expression profile of *Adar* and *Adarb* in adult tissues (Figure 1B) and during embryonic development (Figure 1C,C’). For both paralogs, the highest expression was detected in the neural tube. Noticeably, *Adar* was expressed at a five times higher level than *Adarb* in the neural tube, while they were expressed at similar levels in the gut; moreover, *Adar* but not *Adarb* was also found in the gills. During development, *Adar* and *Adarb* were barely expressed in most of the stages analyzed, with the exception of an expression spike of *Adarb* at 7hpf (hours post fertilization) and the increase of *Adar* expression at pre-metamorphic stages.

### 3.2. ADAR and ADARB Expression through Development by HCR In Situ Hybridisation

We next took advantage of the high sensitivity of the hybridization chain reaction (HCR) method, to detect in situ the low levels of expression of the two Adar genes during the development of *B. lanceolatum,* on two developmental stages: mid-neurula and larva.

In the mid-neurula embryo, *Adar* and *Adarb* (Figure 2A–F) expression was restricted to specific cells in the developing neural tube (Figure 2B–E) and in the proximal half of all somites (Figure 2E,F), except in the first pair. *Adarb* was also expressed throughout the epidermis (best appreciated in Figure 2B). In the larva stage, both *Adar* and *Adarb* (Figure 2G–K) were expressed across the neural tube and in specific clusters of neurons (Figure 2H,I). Outside of the nervous system, *Adar* and *Adarb* transcripts were found in a few cells of the Hatchek’s pit (Figure 2J), in the distal half of the somites (Figure 2L) and in the nuclei of the developing muscle fibers (Figure 2M). In addition, we saw *Adarb* highly expressed in the endostyle (Figure 2J,K).

To gain further insights into the cell types where ADAR-mediated RNA editing might be active, we co-registered both ADAR genes expression with that *of Elav and Prox1*, two genes with key roles in neurogenesis [28,29]

At mid-neurula stage, *Adar* and *Adarb* were co-expressed in the neural tube in *Elav*- and *Prox1*-positive cells (Figure 2C). Moreover, *Adar* and *Prox1* colocalized in the proximal half of the somites (Figure 2E). In the larva *Adar* and *Adarb* were also co-expressed with *Elav* and *Prox1* in the neural tube. However, while in the mid-neurula Adar genes were expressed exclusively in *Elav-* or *Prox1*-positive cells, at the larva stage several *Elav*- and *Prox1*-negative cells express one or both ADAR genes (Figure 2I). Additionally, in the mid-neurula and larva stage *Adar* and *Prox1* were co-expressed in somites and muscles (Figure 2E,M).

### 3.3. A-to-I Editing: SNV Predictions

For each RNA-seq dataset, after looking for single nucleotide variants (SNVs) matching A-to-G (as inosine is detected as a guanosine by synthesis-based sequencing methods) or T-to-C (in the reverse strand) transitions, we filtered them using the genomic reads leaving only the SNVs that were not supported by the genomic reads available at the *B. lanceolatum* genome databases (see methods), hence not perceived as polymorphisms. This yielded a total of 184,163 unique putative RNA-editing events within 23,406 coding loci (from a total of 35,959 coding loci) with a mean 7, 87 SNVs per locus where more than half (109,237 ~59%) would result in a synonymous editing. A complete GO analysis of the genes harboring SNVs can be found in Appendix A. Focusing on tissue specificity, we selected the events appearing exclusively in one or two tissues and plotted them accumulatively relative to each tissue (e.g., tissue1, tissue1 + tissue2, tissue1 + tissue3, etc.). By this filtering, male gonads are the most enriched sample in putative RNA-editing events restricted to one tissue, while the gut is the most enriched overall. According to the expression data, neural tissue would be expected to be highly enriched in putative events, but surprisingly it ranked only in the 6th position. The tissue with the lowest occurrence of putative RNA-editing events was muscle. (Figure 3A). During embryonic development, the number of putative RNA-editing events did not change significantly, with a discrete increase at 10-hpf (Figure 3B).

### 3.4. Orthology-Based Approach to Infer RNA-Editing Conservation

The A-to-I SNV predictions, while being filtered by the genomic reads, could still harbor false positives that were not a result of RNA-editing events, but were due to genomic polymorphisms. We thus performed an orthology-based approach (see methods) to better infer the genuine RNA-editing events, identifying the changes that were conserved between two very distinct species (in this case human and amphioxus). As the homologous amino acid position can vary depending on the quality of the alignment, we implemented a window of homology and tested it at size 0, 1, 5, and infinite amino acids of aligned distance between the two edited amino acids. As the window size increases, so does the permissiveness of the approach, yielding more putative orthologous RNA-editing events. In addition, we separated the events into the following classes: class 0, when the editing resulted in a synonymous change in both interrogated species; class 1, where both the pre-edited and the edited amino acids were conserved between the interrogated species; class 2, where the pre-edited amino acids were not conserved between the interrogated species, but the edited ones were; class 3, where the pre-edited amino acids were conserved between the interrogated species, but the edited ones were not; class 4, where neither the pre-edited nor the edited amino acids were conserved between the interrogated species (Table 1). All the putative orthologous RNA-editing events obtained using the different window sizes except the infinite window size are available in the Appendix A. Interestingly, the tissue specificity for the different window size did not vary significantly except for the infinite window size, with gonads being the most enriched tissues in the other three window sizes (Figure 4).

A detailed list of selected class 1 events at window size 0 is also available (Table 2). We checked manually the quality of the alignments and whether the putative RNA-editing event was in a dsRNA for these events as ADAR proteins act on dsRNA. As a result, 30 events proved to have a good alignment quality with 9 candidates presenting both, a good alignment and the putatively edited nucleotide in a dsRNA region. This shows why a manual curation checking whether the alignments show conservation or have low quality and how is the secondary RNA structure of the gene at the SNV site was hereby needed as there can still be false positive events.

## 4. Discussion

### 4.1. Expression and Conservation of ADAR and ADARB in B. Lanceolatum

The most prominent cases of ADAR-dependent A-to-I editing have been reported in the nervous system and the brain of mammals and other vertebrates. Specifically, editing targets were found in key mediators of synaptic transmission [16]. More recently, a landmark study reported that RNA-editing in cephalopods was extremely abundant in the nervous system, and in particular in genes involved in neuronal excitability and morphology [3]. In mammals, *Adar* knockout mice were shown to be embryonically lethal and mice lacking *Adarb1* died in a few days after birth due to the lack of editing in GluR-B receptor [11,12,30]. According to our expression analysis using a published *B. lanceolatum* RNA-seq dataset [18] (Figure 1B), *Adar* was highly expressed in the neural tube, whereas *Adarb* was found ubiquitously expressed at low levels in a variety of adult tissues. In vertebrates, however, *Adarb1* was expressed more significantly in the neural and muscle tissues, *Adarb2* was predominantly expressed in neural tissue, and *Adar* seemed to have a more pervasive expression [31]. Although a conserved microsyntenic cluster existed between amphioxus and human involving *WDR37* and *ADARB2*, the tissue expression distribution seemed to not be conserved. It was remarkable how from the expected up to four vertebrate paralogs for each gene, *Adar* had only one ortholog without any clear synteny conserved with amphioxus, and *Adarb* had two, but one of them was enzymatically inactive. This could mean that an excess of A-to-I editing can be deleterious as very few organisms had more than two enzymatically active ADAR proteins [32].

The in situ data shown here was the first report of ADAR family expression in amphioxus during development, using the highly sensitive technique of in situ HCR [33], a recent technique still scarcely applied in invertebrates [21,34]. Our results proposed that the amphioxus ADAR family might have a role in gene regulation during the development of the nervous system, as this was where these genes are most prominently expressed. This is congruent with previously reported ADAR-mediated editing in the nervous system of other invertebrates such as *Octopus bimaculoides,* and in vertebrates such as mouse or human. Altogether, this suggests that RNA-editing regulation might have been instrumental for nervous system evolution.

Our results show that *Adar* and *Adarb* were both expressed in the developing neural tube and were initially co-expressed with *Elav* and *Prox1*. Both genes have established roles during neurogenesis: Elav is a well-known pan-neural marker [35] and was active in early stages of neural commitment [36,37], while *Prox1* is a homeobox gene involved in many developmental processes, including neural progenitor differentiation and exit from the cell cycle [29,38,39]. The colocalization of ADAR paralogues with *Elav* and *Prox1* at the mid-neurula stage suggests that RNA editing might play a role in neurogenesis in amphioxus. Interestingly, Takahashi and collaborators [40] demonstrated that the mRNA of *Prox1,* which can act as a tumour suppressor, experienced RNA-editing and loses tumour-suppressive functions in a subset of human cancers. This indicated that in the neural tube and muscles, where ADAR paralogues are co-expressed with *Prox1*, *Adar* may be editing *Prox1*, meaning that the edited isoform of *Prox1* could have a role during the development of these particular tissues and cells.

### 4.2. An Orthology-Based Approach for Highly Polymorphic Species

The first report of an amphioxus genome, already indicated [41] that the amphioxus genome was highly polymorphic. This means that a simple SNV analysis of the RNA-seq reads against the genome, filtering the results for A-to-G and C-to-T (for the RNA-editing in the reverse strand) will not give direct data regarding editing events. A single RNA-seq analysis will result in a mixture of RNA-editing events and the DNA polymorphisms of the samples used for the analysis. One way of reducing the noise caused by the naturally occurring polymorphisms is to find the SNPs present in the genomic reads used to build the genome and filter them out [42], as we did to obtain the data shown in Figure 3. However, not all the polymorphisms actually occurring in an amphioxus population will be present in the available genomic reads, and the RNA-seq and genomic data usually come from different samples, hence some of the putative RNA-editing events could be false positives and be just polymorphisms. To increment the amount of *bona-fide* editing events, we chose to follow an orthology-based approach, taking advantage of the RNA-editing data of a distant but related species, the human [43], and the already reported gene orthology analysis between human and amphioxus [18,34]. Our approach of finding orthologous amino acids (Figure 4, Table 1, see methods) resulted in a strict filter, yielding 1588 putative conserved events. After we implemented a window of homology approach, however, the permissiveness of the pipeline increased and this resulted in 6291 putative events at window size 5, with 405 of them belonging to editing class 1 events (Table 1). An infinite window size was also used to compare edited orthologous pairs without considering whether the edited amino acid was conserved or not. The tissue distribution of the infinite window size (Figure 4D), however, was really similar to the tissue distribution of SNVs (Figure 1A), whereas the three other window sizes had a more similar tissue distribution. As the process is done automatically based on orthologies, a manual curation is needed as shown in Table 2 with the selection of window size 0, class 1 events. Although only nine of these events seem to have had both a good alignment and dsRNA in the edited region, the secondary structure predictions were still quite variable depending on the software used and the length of the sequence interrogated.

Although more analysis must be done to ensure that each of these putative RNA-editing events were not false positives, and problems, like the loss of non-conserved genes in the analysis need to be solved for a proper evolutionary analysis of the RNA-editing, we developed a tool that can be used between any two species to infer putative RNA-editing sites by homology. This method only required that data of RNA-editing exists for the reference species, and genomic and RNA-seq data is needed for the species interrogated.

## 5. Conclusions

We characterized experimentally for the first time the expression of the core RNA editing genes, *Adar* and *Adarb* in *B. lanceolatum* embryos, showing that both were expressed in the neural tube and muscles. We used a genomic filter for countering the polymorphic amphioxus genome and detected 184,163 unique cases of A-to-G and T-to-C variants that are putative RNA-editing events in several adult tissues and during embryonic development. Finally, we developed a pipeline for the assessment of RNA-editing conservation in which the astringency can be adjusted, and widely used with any species having RNA-seq and genomic datasets available.

## Figures and Tables

**Figure 1 genes-11-01440-f001:**
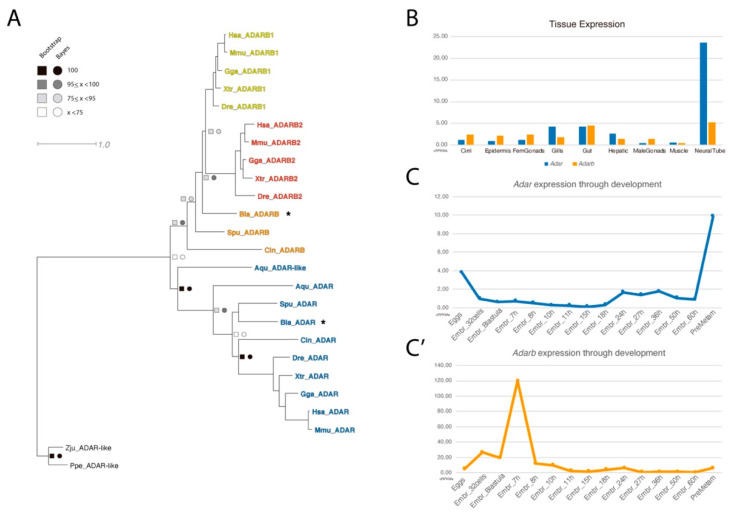
Phylogeny and RNA-seq data of ADAR family. (**A**) Phylogenetic tree of the ADAR family. An asterisk marks the orthologous amphioxus ADAR genes identified, and the scale bar indicates the expected amino acid substitution rate per site. (**B**) Expression of *Adar* and *Adarb1* in adult tissue and (**C**,**C’**) through development, measured in cRPKMs (corrected reads per kilo base per million mapped reads).

**Figure 2 genes-11-01440-f002:**
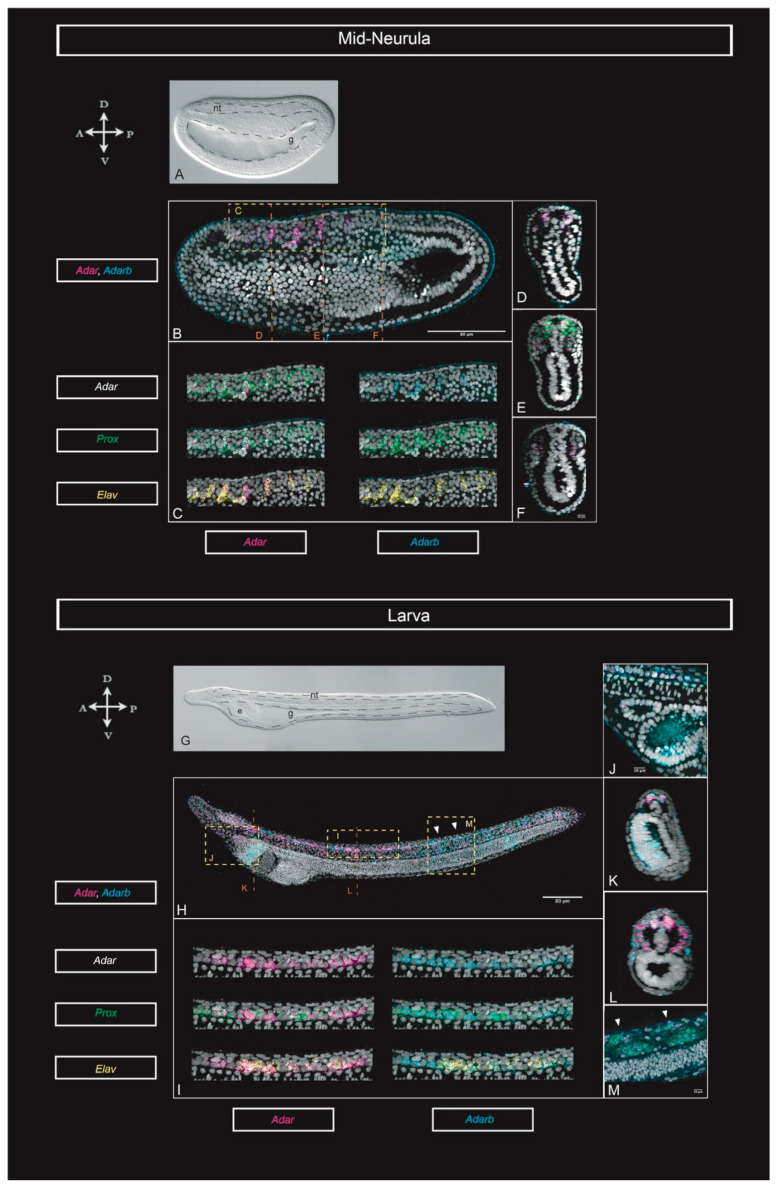
*Adar, Adarb, Elav,* and *Prox1* expression through development. (**A**) Bright field and (**B**) maximum projection for *Adar* and *Adarb*, from quadruple HCR (hybridization chain reaction) in situ hybridizations at the mid-neurula (18 hpf) stage. (**C**) Magnified view from D in B showing the co-expression of *Adar*, *Adarb*, *Prox,* and *Elav* in the neural tube of the mid-neurula. (**D**) Cross section at the level of C in B, showing expression of *Adar* and *Adarb* in the neural tube. (**E**) Cross section at the level of E in B, showing expression of co-expression of *Adar* and *Prox* in the neural tube and somites. (**F**) Cross section at the level of F in B, showing expression of *Adar* and *Adarb* in somites. (**G**) Bright field and (**H**) maximum projection for *Adar* and *Adarb*, from quadruple HCR in situ hybridizations at the early-larva (33 hpf) stage. (**I**) Magnified view of I in H showing the co-expression of *Adar*, *Adarb*, *Prox,* and *Elav* in the neural tube of the early larva. (**J**) Lateral view of the Hatscheck’s pit and the endostyle. (**K**) Cross-section at the level of K in H, showing co-expression of *Adar* and *Adarb* in the neural tube, and expression of *Adarb* in the endostyle. (**L**) Cross-section at the level of L in H showing expression in the neural tube and somites. (**M**) Detail at the level of M in H showing co-expression of *Adar*, *Adarb,* and *Prox* in the lateral muscle fibers. Arrowheads point at the center of the two somites shown. Scale bars: 80 µm and 10 µm for magnified views. Abbreviations: e: endostyle; g: gut; nt: neural tube.

**Figure 3 genes-11-01440-f003:**
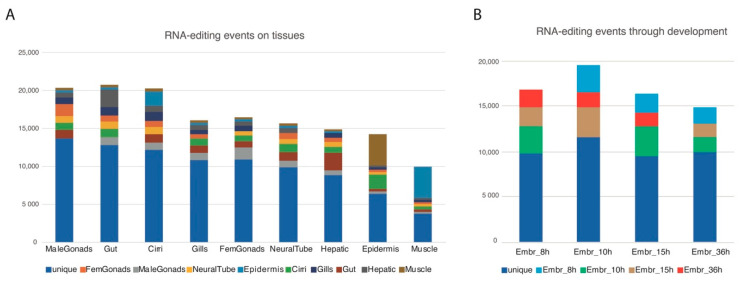
Putative RNA-editing events by tissue and developmental stage. (**A**) Number of putative RNA-editing within coding regions by tissue, specific for one or two of the available tissues. (**B**) Number of putative RNA-editing events within coding loci by developmental stage, specific for one or two of the available stages.

**Figure 4 genes-11-01440-f004:**
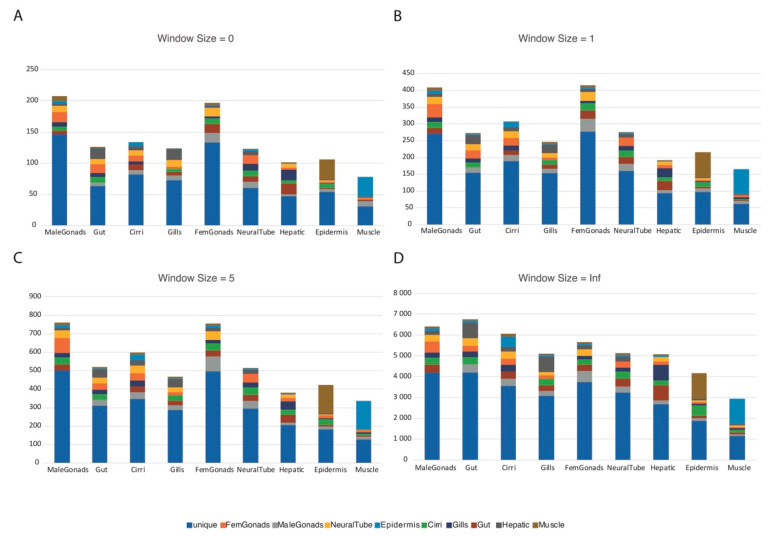
Putative orthologous RNA-editing events by tissue using the tested window sizes. Number of putative orthologous RNA-editing events by tissue, specific for one or two of the available tissues; each graph corresponds to one of the four window sizes analyzed: (**A**) Window size 0; (**B**) Window size = 1; (**C**) Window size = 5; (**D**) Window size = Infinite.

**Table 1 genes-11-01440-t001:** Classification of putative orthologous RNA-editing events within coding loci for each range used. An event can belong to more than one class depending on the isoform analyzed.

	Class 0	Class 1	Class 2	Class 3	Class 4	Total
Window Size = 0	198	177	132	409	875	1588
Window Size = 1	253	230	366	499	2791	3339
Window Size = 5	451	405	1048	915	5920	6291
Window Size = Inf	9814	7476	20,955	16,383	57,075	58,597

**Table 2 genes-11-01440-t002:** Selected windows size 0 class 1 RNA-editing events.

SNV_Coordinates	Tissues	Bla_ID	Hsa_ID	AA_Change	Alignment	Variant in dsRNA	Description
Sc0000141|28964	Embr15h	BL95820	ACTB	K->R	Good	OK	actin and actin-related protein
Sc0000141|28964	Embr15h	BL95820	ACTG1	K->R	Good	OK	actin and actin-related protein
Sc0000141|28935	Embr10h	BL95820	ACTG1	N->D	Good	NO	actin and actin-related protein
Sc0000010|5845185	Epidermis	BL03760	CASP7	Q->R	Good	OK	protease
Sc0000003|746129	Epidermis	BL22951	CD163	N->D	Good	NO	serine protease
Sc0000036|339483	Embr8h	BL13691	CEP295	R->G	Good	NO	Centrosomal protein
Sc0000036|339492	Gills	BL13691	CEP295	T->A	Good	OK	Centrosomal protein
Sc0000169|5290	Embr15h	BL06213	CLN5	T->A	Good	NO	Ceroid-Lipofuscinosis Neuronal Protein
Sc0000106|847254	Cirri; Embr8h	BL04964	CNTN6	K->R	Good	NO	scaffold/adaptor protein
Sc0000013|2542412	Hepatic	BL22946	CYP3A43	K->R	Good	NO	oxygenase
Sc0000017|3071594	Cirri; Embr36h; Hepatic; MaleGonads	BL14540	ERAP1	I->V	Good	NO	metalloprotease
Sc0000170|219193	Cirri	BL20200	FBXO11	H->R	Good	NO	ubiquitin-protein ligase
Sc0000176|48786	Embr15h	BL03944	GNL3	K->E	Good	NO	GTPase activity
Sc0000644|3600	NeuralTube	BL15637	GRIK2	I->V	Good	OK	Glutamate receptor
Sc0000049|546225	Epidermis	BL12702	IFT81	X->W	Good	NO	Intraflagellar transport protein
Sc0000154|605189	Cirri;Hepatic	BL95836	IGFBP7	K->E	Good	NO	insulin-like growth factor binding
Sc0000038|1503269	Embr8h; Embr15h	BL03525	KLHL29	H->R	Good	NO	scaffold/adaptor protein
Sc0000054|734626	Embr8h	BL06611	MAP4K4	T->A	Good	NO	transferase activity
Sc0000021|2537001	Cirri; Embr36h	BL23627	MICAL3	Q->R	Good	NO	actin binding and Rab GTPase binding
Sc0000176|396675	Gut	BL74980	NUDT6	I->V	Good	NO	nucleotide phosphatase
xfSc0000677|1518	Cirri; Embr8h; Embr10h; Epidermis;FemGonads;Gills;Gut; Hepatic;MaleGonads	BL09048	NUDT6	K->R	Good	OK	nucleotide phosphatase
Sc0000080|164844	NeuralTube	BL07116	OXSM	I->V	Good	NO	3-oxoacyl-[acyl-carrier-protein] synthase activity
Sc0000027|1956570	Embr8h; Embr10h; Embr15h; Epidermis; FemGonads; NeuralTube	BL54853	PDHA1	K->R	Good	NO	oxidoreductase activity
Sc0000000|9845821	Cirri; Embr8h; Embr10h; Embr15h; Embr36h; Epidermis; FemGonads; Gut; Hepatic; NeuralTube	BL07590	PRIMPOL	Y->C	Good	NO	DNA-directed DNA polymerase activity
Sc0000312|38338	Embr10h	BL18204	PSIP1	K->E	Good	OK	transcription cofactor
Sc0000100|854356	NeuralTube	BL07716	RPS5	K->R	Good	OK	ribosomal protein
Sc0000029|517144	Gills; Gut	BL02932	SLC35A3	X->W	Good	OK	transporter
Sc0000017|3058040	Gut	BL14540	TRHDE	N->S	Good	NO	metalloprotease
Sc0000119|132511	Gills	BL04457	TUBA1B	K->E	Good	NO	tubulin

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
