# Peer review of "The ADAR Family in Amphioxus: RNA Editing and Conserved Orthologous Site Predictions"

_genes, 2020, doi:10.3390/genes11121440_

Round 1

Reviewer 1 Report

The study by Zawisza-Álvarez et al. 'The Adar family in amphioxus: RNA editing at the I/V border and conserved orthologous site predictions' is a pioneer work investigating RNA editing in the cephalochordate Branchiostoma lanceolatum. In the first part of the manuscript, the authors describe identification of the two members of the Adar family and their expression using published RNA-seq data and HCR in situ hybridization. In the second part, the authors present a method to tentatively identify editing sites in coding sequences by filtrering genomic SNPs out of SNVs from RNA-seq data. Finally, the authors propose that comparison with human data should help refining relevant sites and identify two orthologous sites. While this study has strong potential in the poorly studied field of evolution of RNA editing, several concerns have to be addressed before providing sufficient new insights.

Generalities about RNA editing.

A focus is set on the role of editing in nervous system complexification in vertebrates and cephalopds. The authors should describe the extent of editing, the number and types of genes, but also whether this occurs during embryogenesis or adult life. It would important to describe whether editing in the nervous system is conserved among vertebrates but also between vertebrates and cephalopods.

Adar-editing is also involved in viral RNA editing. This should be mentioned together with other functions of editing.

Is there RNA editing that does not involve the Adar gene family?

Expression of Adar genes in amphioxus.

While this may be the result of compression for making a pdf file, the quality of the images of HCR ISH is very poor. Beside CNS expression, the reader is left to trust the description in the text to learn gene expression. The images are very small in particular for larvae.

To 'save' space and enlarge the figure, the authors could remove Prox1 that is somewhat redundant with Elav.

It is a pity that the authors do not show ISH at key time points that are revealed by the RNA-seq data: 7hrs, premetamorphosis and adut (neural tube).

The textual description of the patterns is poor and not specific enough. For example, 'medial portion of all somites' (l133) may mean along the medio-lateral axis, but most probably along the antero-posterior axis according to Fig2A. Similar for 'mid-section' (l137)… Also Adarb1 seems to be expressed in the entire epidermis, not only in the anterior part (Fig2B).

Identification of putative editing sites in amphioxus.

While I understand that the almost 259.000 sites in the coding loci are an overestimation of actual editing since many correspond to polymorphism, I find the analysis quite limited. It would be interesting to further describe the number of genes that correspond to these sites, the number of SNVs per gene, percentage of genes in the genome harboring SNVs, silent SNVs, GO analysis…

Evolutionary conservation of editing sites with human.

The authors chose to reveal relevant SNVs using orthology with human. While this is an interesting approach, it is not clear whether it is relevant. For example, the authors do not provide information on the conservation of editing in vertebrates. Are the same genes edited, and at the same position? If not, the authors might consider orthologous genes, not orthologous sites, but sites whose editing lead to changes in amino acid sequence.

Their approach identifies a limited number of fully conserved sites. Does it mean that RNA editing is not conserved between human and amphioxus. The biological role of these genes is not described.

I would suggest the authors to use available genomic and transcriptomic data from other Branchiosotoma species. Applying for each species the same approach SNVs identification/SNPs filtrering, then considering orthologous filtered SNVs, this seems more likely to identity relevant candidate editing sites than orthology with the distant human genome.

Discussion.

The authors describe potentially broad distributions of editing in amphioxus with a marked prevalence in gonads. Yet their discussion is limited to nervous system.

Minor points:

- I/V likely means invertebrate/vertebrate. This abbreviation does not seem that common and should be avoided in a title.

- l88: the human RNA editing database is not described nor referenced.

- the phylogenetic tree indicates that the so-called BlaAdarB1 is orthologous to both AdarB1 and AdarB2 from vertebrates. This gene should thus be named AdarB or AdarB1/2.

- explain the importance of dsRNA (Table2).

- l242/243: I do not understand the logic of conservation of Adar editing in nervous system between amphioxus and vertebrates, and this being 'instrumental' in nervous sytem evolution and at the invertebrate/vertebrate transition.

Author Response

Comments and Suggestions for Authors

Referee 1

The study by Zawisza-Álvarez et al. 'The Adar family in amphioxus: RNA editing at the I/V border and conserved orthologous site predictions' is a pioneer work investigating RNA editing in the cephalochordate Branchiostoma lanceolatum. In the first part of the manuscript, the authors describe identification of the two members of the Adar family and their expression using published RNA-seq data and HCR in situ hybridization. In the second part, the authors present a method to tentatively identify editing sites in coding sequences by filtrering genomic SNPs out of SNVs from RNA-seq data. Finally, the authors propose that comparison with human data should help refining relevant sites and identify two orthologous sites. While this study has strong potential in the poorly studied field of evolution of RNA editing, several concerns have to be addressed before providing sufficient new insights.

Generalities about RNA editing.

A focus is set on the role of editing in nervous system complexification in vertebrates and cephalopds. The authors should describe the extent of editing, the number and types of genes, but also whether this occurs during embryogenesis or adult life. It would important to describe whether editing in the nervous system is conserved among vertebrates but also between vertebrates and cephalopods.

We have added some examples of editing not mediated by ADAR (lines 37-40) as well as some bibliography about the conservation of editing in vertebrates (49-50). Unfortunately, the conservation of editing in the nervous system between distant clades as vertebrates and cephalopods has not been studied enough. A future work could be to run our pipeline using human and octopus to check the editing conservation between them.

Adar-editing is also involved in viral RNA editing. This should be mentioned together with other functions of editing.

We have added (45-49) information about the relationship between the Adar family and the viral genomes.

Is there RNA editing that does not involve the Adar gene family?

 Yes, excuses, we have included the info in the manuscript (37-40)

Expression of Adar genes in amphioxus.

While this may be the result of compression for making a pdf file, the quality of the images of HCR ISH is very poor. Beside CNS expression, the reader is left to trust the description in the text to learn gene expression. The images are very small in particular for larvae.

We share the concern of the reviewer about the pdf compression reducing the quality of the images, so together with the all-inclusive revised pdf of the manuscript, we uploaded a high-resolution version of the revised Fig. 2.

For further clarity on the images, we also followed the advice of the reviewer on the size and detail of the figure. Accordingly, the revised Fig. 2 now includes bigger images of the whole embryos and several panels with close-up views showing in detail the expression in the CNS.

To 'save' space and enlarge the figure, the authors could remove Prox1 that is somewhat redundant with Elav.

We have also followed the recommendation of the reviewer in this point and we have removed the dedicated panels to Prox and Elav alone. Instead, we now show them only in combination with Adar and Adarb (Please see Fig. 2C,E, I, M).

It is a pity that the authors do not show ISH at key time points that are revealed by the RNA-seq data: 7hrs, premetamorphosis and adut (neural tube).

We agree with the referee that this is a pity. Its absence is due to the pandemic coronavirus limitations of fieldwork to obtain embryos at this particular stage, which is absent from our stocks. We centered on available embryonic stages, but we agree that further work will include pre- and post-metamorphosis and adult tissues

The textual description of the patterns is poor and not specific enough. For example, 'medial portion of all somites' (l133) may mean along the medio-lateral axis, but most probably along the antero-posterior axis according to Fig2A. Similar for 'mid-section' (l137)… Also Adarb1 seems to be expressed in the entire epidermis, not only in the anterior part (Fig2B).

We have further detailed the description of the pattern both in the figure legend (162-176) and in the main text (152-159;181-187). To help the description of the patterns we have also added specific panels in Fig. 2 showing magnified views of the structure described every time.

We agree with the reviewer that 'medial portion of all somites' was not accurate enough, so we have rephrased as follows: ‘proximal half of all somites’ and we have accompanied the statement by making reference to Fig. 2D-F, where we show cross-sections at the level of the somites.

The phrase containing ‘Mid-section’ has been replaced, as we thought it was not accurate enough. As for the epidermal expression of Adarb, we have gone through the expression of several embryos at this stage and we noticed the reviewer was right, so we have rephrased as follows: "Adarb is also expressed throughout the epidermis (best appreciated in Fig. 2B).” (154)

Identification of putative editing sites in amphioxus.

While I understand that the almost 259.000 sites in the coding loci are an overestimation of actual editing since many correspond to polymorphism, I find the analysis quite limited. It would be interesting to further describe the number of genes that correspond to these sites, the number of SNVs per gene, percentage of genes in the genome harboring SNVs, silent SNVs, GO analysis…

We have rerun our analysis using the latest available databases and the total SNVs is now 184.163 (194). We agree with the reviewer about the missing info on the SNVs and we have added it to the text (194-197). We also run a GO analysis which data can be seen in the new  Fig. S1.

Evolutionary conservation of editing sites with human.

The authors chose to reveal relevant SNVs using orthology with human. While this is an interesting approach, it is not clear whether it is relevant. For example, the authors do not provide information on the conservation of editing in vertebrates. Are the same genes edited, and at the same position? If not, the authors might consider orthologous genes, not orthologous sites, but sites whose editing lead to changes in amino acid sequence.

We have included bibliography in the introduction about the existence of conserved edited amino acids in vertebrates (49-50). We think that looking into orthologous genes without considering whether the edited amino acid is conserved or not may give useful insights. Thus, we implemented the option to run the pipeline with an infinite window size. The results can be seen in Table 1 and Fig. 4D.

Their approach identifies a limited number of fully conserved sites. Does it mean that RNA editing is not conserved between human and amphioxus. The biological role of these genes is not described.

The number of fully conserved sites has increased drastically when using the latest databases. In Table 2 now we show the 30 orthologus SNVs with a good alignment. In addition, the profile of the tissue expression distribution (Fig. 4) is substantially different in the window sizes 0,1 and 5 to the one shown for all the SNVs (Fig. 3A), while the one from the infinite window size (Fig. 4D) is more similar to the profile seen in all the SNVs. This could indicate that, while a huge amount of SNPs still populate the SNVs and infinite window size data, the orthology filter is effective. We also added to the Table 2 some info about the biological role of those genes identified.

I would suggest the authors to use available genomic and transcriptomic data from other Branchiosotoma species. Applying for each species the same approach SNVs identification/SNPs filtrering, then considering orthologous filtered SNVs, this seems more likely to identity relevant candidate editing sites than orthology with the distant human genome.

While we agree with the reviewer that this could provide us of useful insights, we could not perform the analysis in the brief time granted for the revisions. Besides, the RNA editing data from human is populated with bona fide events making it easier for comparisons. 

Discussion.

The authors describe potentially broad distributions of editing in amphioxus with a marked prevalence in gonads. Yet their discussion is limited to nervous system.

Although the conserved editing events seem to have a marked prevalence in gonads, our expression data from the available adult tissues and the ish data indicate that there is an important expression of both Adar genes in the nervous system. It could be that the editing per se is more abundant in the nervous system in a relatively small set of genes, while a greater set of genes is edited less frequently in gonads.

Minor points:

- I/V likely means invertebrate/vertebrate. This abbreviation does not seem that common and should be avoided in a title.

Agreed

- l88: the human RNA editing database is not described nor referenced.

Referenced in methods (108)

- the phylogenetic tree indicates that the so-called BlaAdarB1 is orthologous to both AdarB1 and AdarB2 from vertebrates. This gene should thus be named AdarB or AdarB1/2.

Agreed. We changed it to Adarb through the figures and text.

- explain the importance of dsRNA (Table2).

We added a brief explanation (248)

- l242/243: I do not understand the logic of conservation of Adar editing in nervous system between amphioxus and vertebrates, and this being 'instrumental' in nervous sytem evolution and at the invertebrate/vertebrate transition.

We have rephrased that sentence to “Altogether, this suggests that RNA-editing regulation might have been instrumental for nervous system evolution” (278-279)

Reviewer 2 Report

The authors focus on the Adar genes of amphioxus as a means to better understand the role of RNA editing in evolution, aiming to specifically focus on the invertebrate-vertebrate transition. In addition to a range of new expression data for these genes in amphioxus the authors also provide a new bioinformatic pipeline for detecting RNA editing in polymorphic species, which should facilitate inter-species comparisons and hence greatly improve our ability to study RNA editing more widely and in a clearly evolutionary context.

Major revisions.

1) From the phylogenetic tree shown in Fig.1 it looks as though the authors have not named one of their amphioxus genes correctly. The gene currently named as Bla_ADARB1 is actually pro-orthologous to vertebrate ADARB1 and B2 genes. As such the amphioxus gene should be named something like Bla_ADARB1/2. This of course is presuming that the tree is rooted with the Zju_ADAR-like and Ppe_ADAR-like genes, but this is not clear due to the insufficient description of how the tree was constructed. The way that the tree was constructed should be described either in the Methods or in the figure legend, and the accession number and alignment for all of the genes in this tree must also be provided.

2) Perhaps it is also noteworthy that vertebrates look to have undergone plenty of loss of ADAR genes, as one might expect upto four vertebrate genes for each of the ADAR and ADARB1/2 clades. If the authors are interested in these patterns of gene duplication and loss then they would need to examine the synteny data around these chordate ADAR genes. If the authors are not interested in establishing these precise evolutionary patterns then they should at least briefly comment on these possibilities in their discussion, as these evolutionary events might be significant for the evolution of gene functions, particularly in vertebrates.

3) The expression detailed in section 3.2 is difficult to see in Fig.2 in several respects.

- Adarb1 expression in mid-neurula neural tube is very weak and only really observable in the figure if it is greatly magnified by the reader (and some generous imagination used). Perhaps the authors could make this easier for the readers by providing a larger panel of (at least) the mid-neurula Adarb1 data.

- It seems odd to say Adar is strongly expressed in the most anterior epidermis when there appears to be staining all around the entire epidermis, and if anything it is the posterior epidermis that stains more strongly.

- In the larvae, Adar is said to be expressed in developing muscle fibres. This expression, if it is there, is not visible in the figure.

- Also, I cannot see the expression of Adarb1 in the endostyle.

- Given these shortcomings in the detail visible in Fig.2 then it is similarly difficult to see the sorts of expression and co-expression described for the Adar genes and Elav and Prox1 as well.

4) Some elements of the Discussion do not make scientific sense.

Firstly, since the two main clades of Adar genes (Adar and Adarb1/2) both have amphioxus (and other invertebrate) members then why do the authors hypothesize that the whole genome duplication events at the origin of vertebrates might have led to some sort of swapping event between the regulation of these two types of gene? The duplication to produce Adar and Adarb1/2 had clearly already happened well before the vertebrate WGD.

5) Secondly, why hypothesize that RNA-editing in neural tissues is key at the invertebrate-vertebrate transition? The authors provide evidence that RNA-editing is likely to be occurring in the amphioxus nervous system, and it is already known to occur in the octopus nervous system. Thus, for the authors to speculate a role for RNA-editing in the elaboration of the vertebrate nervous system then they would need to be providing evidence for more extensive RNA-editing in vertebrate CNSs relative to invertebrates, or editing of genes that are essential for vertebrate CNS development, but which are not edited in invertebrates.

6) This second point then relates to the title and raises the question as to whether the invertebrate-vertebrate transition should be mentioned in the title at all, as it seems at present as though nothing much at all is being said about this trnasition.

7) There is another problem with the hypothesis about expression swapping between amphioxus and humans. Amphioxus Adarb1/1 is pro-orthologous to human ADARB1 and B2 according to the authors phylogenetic tree. Therefore, any comparisons of expression should include both human genes, not only ADARB1.

8) A further reason why the swapping hypothesis does not make sense is that the authors show that both Adar and Adarb1 in amphioxus are expressed in a variety of adult tissues (Fig.1). Thus, there is not such a stark contrast between amphioxus Adar and Adarb1 and the human orthologues as the authors try to portray.

On these grounds I think that the expression swapping hypothesis should be removed from this manuscript as the data do not support it.

Minor revisions

9) Title: “the I/V border” is not a particularly clear term to use. I suggest changing this to, “the invertebrate-vertebrate transition” (although this should probably be removed completely – see comment above).

10) Abstract, line 15: change “vastly” to “extensively”.

11) Abstract, line 17: I am not quite sure what the authors mean by “most of the clades”. Is the meaning ‘most other clades’ (i.e. animals besides cephalopods and mammals)? Or are the authors referring to animal clades in general? If it is this second meaning then the phrase “in most of the clades” can simply be deleted.

12) Abstract, line 19: it is not entirely accurate to say that amphioxus ‘represents’ the invertebrate-vertebrate transition. Rather, it is at a key phylogenetic position that provides insight into the invertebrate-vertebrate transition.

13) Introduction, line 29: change “holders” to “possessors”.

14) Introduction, line 32: change “and adding new genes” to “by addition of extra genes”.

15) Introduction, line 38: change “aminoacidic” to “amino acid”. See also line 94. See also Fig.1 legend.

16) Introduction, line 53: change “genome” to “genomic DNA”, and change “enough” to “sufficient”.

17) Introduction, line 59: change “development on” to “development of”.

18) Methods, line 99: insert a space between ‘amino acids’. Also on this line, the sentence ‘we also filtered…’ as written makes it sound as though this filtering against the candidate SNPs was done twice; once in the initial generation of the candidate list described in the previous Methods subsections and then again at this point in the pipeline. I presume that this filtering was only done once, and this was at this point in the pipeline, as described on line 99? If this is correct then the writing could be clarified here (e.g. change “We also filtered…” to “It was at this point that we filtered the results with the confirmed SNPs as described above”).

19) Results, line 108: change the subheading to either “Amphioxus have two Adar genes” or “The European amphioxus has two Adar genes”. The second option is more accurate if the authors have not examined the Adar complement of other amphioxus species besides lanceolatum.

20) Fig.1 legend, line 124: change “orthologs” to “orthologous”. And line 126, change “trough” to “through”.

21) Results, line 168” typo with “by the by”?

22) Results, line 169: add an ‘s’ to give “SNVs”.

23) Results, line 173 (and in Fig.3 legend): what is meant by “exclusively in one or two tissues”. My reading of Fig.3 is that many of the SNVs are expressed in multiple tissues, not only one or two, judging from colours representing specific tissues appearing in nearly all of the other tissue types assessed.

24) Section 3.3: are these numbers without the filtering against the human orthologues? This is implied to be the case from reading section 3.4, but the Methods section implied that the filtering against human orthologues was done for the entirety of the analyses. This needs to be clearer in the manuscript (perhaps by clarifying the diversity of analyses done by rewording the Methods).

25) The methods are similarly unclear in connection with Table 2. The Methods section describes the filtering against whether the SNV is in potential dsRNA, but then Table 2 implies that a number of the SNVs analysed are not in dsRNA.

Both of these issues make it very difficult for the reader to understand which components of the pipeline were used to produce the numbers in the figures.

26) Results Section 3.4: insert a space to give “amino acids” in multiple places in this paragraph.

27) Results, line 192: what is meant by ‘range of homology’? I think what the authors mean is that they implemented several different amino acid window sizes within their alignments in which they would call a potential editing event as at a homologous location.

28) Results, line 219: what is meant by “manual curation of the data”? What does this curation involve? Are the authors simply manually editing the alignments (and if so then simply say this) or does this curation involve something else? It would help if further information was given here on how much this manual curation improved the estimates of RNA-editing. (From my reading of the later part of the Discussion I now presume that ‘manual curation’ simply means that the alignment is deemed to be good and that the variant is in dsRNA – if this is the case then this needs to be spelled out in the results. However, in what sense is this ‘manual curation’, since both the alignments and dsRNA prediction are done by a computer?).

29) Discussion, line 232: change “it is Adarb1 the gene expressed” to “it is the Adarb1 gene that is expressed” (but also note the incorrect gene name here as well, as highlighted in point 1 above).

30) Discussion, line 235: the word ‘earthquakes’ is not appropriate here. Change to something like ‘innovations’ or ‘elaborations’.

31) Discussion, line 252: change “suffers” to “experiences”.

32) Discussion, line 260: it is not appropriate to talk of ‘the’ amphioxus genome when actually several different amphioxus species have been sequenced and the authors are referring to a different species to the one that they have analysed in this manuscript. Thus, line 260 should be changed to something like, “The first report of an amphioxus genome, already indicated [32] that amphioxus genomes are highly polymorphic”.

33) Discussion, line 270: change “choose” to “chose”.

34) Discussion, line 273: space between “amino acids”.

35) Discussion, line 276: add an ‘s’ to “events” in “events (Table 1)”.

36) Conclusion, lines 293-4: change “which astringency that can be finely adjusted” to something like “in which the stringency can be adjusted”.

37) Queries about the pipeline…

In order to remove SNPs from the candidate SNV events the authors use the polymorphism data available in the lanceolatum genome sequence. Is this polymorphism data sufficient to do a good job of removing a high proportion of lanceolatum SNPs? That is, a population-level estimate of SNPs will presumably be much more extensive than the number of SNPs obtained from the deliberately limited number of SNPs produced from sequencing a single diploid individual to obtain the lanceolatum draft genome sequence. Although the authors allude to this in their discussion, it could be more explicitly highlighted and commented on, in terms of how this approach could be improved upon in future.

38) What is the availability of pipeline code?

39) One limitation of the complete pipeline is that it focuses only on editing events that are similar (and potentially homologous?) to events found in human genes. RNA editing that occurs in a particular species more generally will not be assessed. This is unfortunate from an evolutionary perspective, as there will obviously be much evolution occurring via genes that either do not have human homologues, or can be edited at different locations from the human homologue. This aspect of the work could also be briefly discussed.

Perhaps (for future work?) the authors could run the pipeline by searching for octopus homologues that are edited alongside the human-focused analyses done here to help increase confidence in RNA-editing events that might be conserved across octopus, amphioxus and humans, and to provide a complimentary dataset of octopus-amphioxus events versus the amphioxus-human events.

Author Response

Comments and Suggestions for Authors

Referee 2

The authors focus on the Adar genes of amphioxus as a means to better understand the role of RNA editing in evolution, aiming to specifically focus on the invertebrate-vertebrate transition. In addition to a range of new expression data for these genes in amphioxus the authors also provide a new bioinformatic pipeline for detecting RNA editing in polymorphic species, which should facilitate inter-species comparisons and hence greatly improve our ability to study RNA editing more widely and in a clearly evolutionary context.

Major revisions.

 First of all, I would like to address the differences in the data from the previous version of the manuscript. To comply with one of the suggestions of Reviewer 1 (the infinite window size), we had to update our databases. This resulted in a reduction of the SNVs in amphioxus but came with an increment in the orthologous SNVs found.

1) From the phylogenetic tree shown in Fig.1 it looks as though the authors have not named one of their amphioxus genes correctly. The gene currently named as Bla_ADARB1 is actually pro-orthologous to vertebrate ADARB1 and B2 genes. As such the amphioxus gene should be named something like Bla_ADARB1/2. This of course is presuming that the tree is rooted with the Zju_ADAR-like and Ppe_ADAR-like genes, but this is not clear due to the insufficient description of how the tree was constructed. The way that the tree was constructed should be described either in the Methods or in the figure legend, and the accession number and alignment for all of the genes in this tree must also be provided.

We have renamed it to Adarb through the whole manuscript, as cleaverly suggested by the reviewer. In addition, we have added a new section in material and methods about the generation of the phylogenetic tree including the accession numbers of the sequences used (71-79).

2) Perhaps it is also noteworthy that vertebrates look to have undergone plenty of loss of ADAR genes, as one might expect upto four vertebrate genes for each of the ADAR and ADARB1/2 clades. If the authors are interested in these patterns of gene duplication and loss then they would need to examine the synteny data around these chordate ADAR genes. If the authors are not interested in establishing these precise evolutionary patterns then they should at least briefly comment on these possibilities in their discussion, as these evolutionary events might be significant for the evolution of gene functions, particularly in vertebrates.

We have now discussed this in the corresponding section (266-271)

3) The expression detailed in section 3.2 is difficult to see in Fig.2 in several respects.

- Adarb1 expression in mid-neurula neural tube is very weak and only really observable in the figure if it is greatly magnified by the reader (and some generous imagination used). Perhaps the authors could make this easier for the readers by providing a larger panel of (at least) the mid-neurula Adarb1 data.

We understand the images in the figure were too small to see the details we described in the text, spacially after pdf compression, hence revised Fig. 2 now includes bigger images of the whole embryos and several panels with close-up views showing the details we describe in the text.

- It seems odd to say Adar is strongly expressed in the most anterior epidermis when there appears to be staining all around the entire epidermis, and if anything it is the posterior epidermis that stains more strongly.

Having noted the observation of the reviewer, we checked through the stacks of the different embryos hybridized at this stage and found that indeed Adarb1 is expressed throughout the epidermis at the neurula stage, so we have corrected this, thank you

- In the larvae, Adar is said to be expressed in developing muscle fibres. This expression, if it is there, is not visible in the figure.

We have added a panel: Fig.2M, showing a magnified view of this.

- Also, I cannot see the expression of Adarb1 in the endostyle.

We have added two panels: Fig.2J and K , showing a magnified view of both Adar and Adarb1 in the endostyle.

- Given these shortcomings in the detail visible in Fig.2 then it is similarly difficult to see the sorts of expression and co-expression described for the Adar genes and Elav and Prox1 as well.

To address this, we have added magnified views showing: 1. Co-expression between Adar and Adarb1: Fig.2B, D, F, H-L; and 2. Co-expression between Adar or Adarb1 with Elav or Prox: Fig. 2C, E, I, M.

4) Some elements of the Discussion do not make scientific sense.

Firstly, since the two main clades of Adar genes (Adar and Adarb1/2) both have amphioxus (and other invertebrate) members then why do the authors hypothesize that the whole genome duplication events at the origin of vertebrates might have led to some sort of swapping event between the regulation of these two types of gene? The duplication to produce Adar and Adarb1/2 had clearly already happened well before the vertebrate WGD.

The reviewer affirmation is correct, and we have deleted the swapping hypothesis (267-270)

5) Secondly, why hypothesize that RNA-editing in neural tissues is key at the invertebrate-vertebrate transition? The authors provide evidence that RNA-editing is likely to be occurring in the amphioxus nervous system, and it is already known to occur in the octopus nervous system. Thus, for the authors to speculate a role for RNA-editing in the elaboration of the vertebrate nervous system then they would need to be providing evidence for more extensive RNA-editing in vertebrate CNSs relative to invertebrates, or editing of genes that are essential for vertebrate CNS development, but which are not edited in invertebrates.

We have removed the invertebrate-vertebrate transition sentence. We would like to gain insights on the different levels of RNA-editing between the invertebrate and vertebrate CNSs, but as for now  this is out of the scope of this paper.

6) This second point then relates to the title and raises the question as to whether the invertebrate-vertebrate transition should be mentioned in the title at all, as it seems at present as though nothing much at all is being said about this trnasition.

We have changed the title accordingly.

7) There is another problem with the hypothesis about expression swapping between amphioxus and humans. Amphioxus Adarb1/1 is pro-orthologous to human ADARB1 and B2 according to the authors phylogenetic tree. Therefore, any comparisons of expression should include both human genes, not only ADARB1.

We have now included the three Adar genes in the comparisons (262-271)

8) A further reason why the swapping hypothesis does not make sense is that the authors show that both Adar and Adarb1 in amphioxus are expressed in a variety of adult tissues (Fig.1). Thus, there is not such a stark contrast between amphioxus Adar and Adarb1 and the human orthologues as the authors try to portray.

The swapping theory has been removed.

On these grounds I think that the expression swapping hypothesis should be removed from this manuscript as the data do not support it.

We agreed. The swapping hypothesis has been removed as there is not enough data to support it.

Minor revisions

9) Title: “the I/V border” is not a particularly clear term to use. I suggest changing this to, “the invertebrate-vertebrate transition” (although this should probably be removed completely – see comment above).

We have removed the I/V portion of the title.

10) Abstract, line 15: change “vastly” to “extensively”.

 Changed

11) Abstract, line 17: I am not quite sure what the authors mean by “most of the clades”. Is the meaning ‘most other clades’ (i.e. animals besides cephalopods and mammals)? Or are the authors referring to animal clades in general? If it is this second meaning then the phrase “in most of the clades” can simply be deleted.

It is the first option, we have clarified now (16).

12) Abstract, line 19: it is not entirely accurate to say that amphioxus ‘represents’ the invertebrate-vertebrate transition. Rather, it is at a key phylogenetic position that provides insight into the invertebrate-vertebrate transition.

 Changed (18).

13) Introduction, line 29: change “holders” to “possessors”.

 Changed (28).

14) Introduction, line 32: change “and adding new genes” to “by addition of extra genes”.

 Changed (31).

15) Introduction, line 38: change “aminoacidic” to “amino acid”. See also line 94. See also Fig.1 legend.

 Aminoacidic and aminoacid changed to amino acid though all the manuscript.

16) Introduction, line 53: change “genome” to “genomic DNA”, and change “enough” to “sufficient”.

 Changed (57).

17) Introduction, line 59: change “development on” to “development of”.

 Changed (64).

18) Methods, line 99: insert a space between ‘amino acids’. Also on this line, the sentence ‘we also filtered…’ as written makes it sound as though this filtering against the candidate SNPs was done twice; once in the initial generation of the candidate list described in the previous Methods subsections and then again at this point in the pipeline. I presume that this filtering was only done once, and this was at this point in the pipeline, as described on line 99? If this is correct then the writing could be clarified here (e.g. change “We also filtered…” to “It was at this point that we filtered the results with the confirmed SNPs as described above”).

 Aminoacidic and aminoacid changed to amino acid though all the manuscript. Regarding the filtering, the reviewer is correct. We filter the transcripts only once, and the second description has been removed.

19) Results, line 108: change the subheading to either “Amphioxus have two Adar genes” or “The European amphioxus has two Adar genes”. The second option is more accurate if the authors have not examined the Adar complement of other amphioxus species besides lanceolatum.

Subheading changed (128).

20) Fig.1 legend, line 124: change “orthologs” to “orthologous”. And line 126, change “trough” to “through”.

Both changed (144;146)

21) Results, line 168” typo with “by the by”?

Correct, changed to just “by” (191).

22) Results, line 169: add an ‘s’ to give “SNVs”.

Changed (192).

23) Results, line 173 (and in Fig.3 legend): what is meant by “exclusively in one or two tissues”. My reading of Fig.3 is that many of the SNVs are expressed in multiple tissues, not only one or two, judging from colours representing specific tissues appearing in nearly all of the other tissue types assessed.

The figure is composed with the SNVs expressed in one or two tissues and represented relative for each tissue. For example, there may be 23 SNVs found exclusively in Cirri and 12 found in Cirri and Hepatic and 5 found in Cirri and Muscle. The representation when focus on cirri will be a 23 sized “blue” column (unique) with a 12 sized “purple” segment on top of the “blue” (cirri+hepatic) tipped by a 10 sized “dark gold” segment (cirri+muscle). A small clarification has been added to the text to help the reader interpret the figure (199).

24) Section 3.3: are these numbers without the filtering against the human orthologues? This is implied to be the case from reading section 3.4, but the Methods section implied that the filtering against human orthologues was done for the entirety of the analyses. This needs to be clearer in the manuscript (perhaps by clarifying the diversity of analyses done by rewording the Methods).

The reviewer is correct. The section 3.3 data is just the SNV found using the third and fourth sections of methods (87-100). The human orthologues analysis was done for the entire of the analyses but presented and discussed separately. We have made some changes to clarify this.

25) The methods are similarly unclear in connection with Table 2. The Methods section describes the filtering against whether the SNV is in potential dsRNA, but then Table 2 implies that a number of the SNVs analysed are not in dsRNA.

 The dsRNA section of the methods describes part of what we have referred as “manual curation”. The secondary structure is not a part of the pipeline per se and the events can be on dsRNA or not. That is why some of the Table 2 events are not in dsRNA. We have made some changes in the methods section to clarify this.

Both of these issues make it very difficult for the reader to understand which components of the pipeline were used to produce the numbers in the figures.

We apologize for the lack of clarity regarding which components of the pipeline were used to produce the figures and hope that the changes made through the manuscript help the reader to understand better to which analysis belongs each output.

26) Results Section 3.4: insert a space to give “amino acids” in multiple places in this paragraph.

Aminoacidic and aminoacid changed to amino acid though all the manuscript. Regarding the filtering, the reviewer is correct. We filter the transcripts only once, and the second description has been removed.

27) Results, line 192: what is meant by ‘range of homology’? I think what the authors mean is that they implemented several different amino acid window sizes within their alignments in which they would call a potential editing event as at a homologous location.

The reviewer is correct and we have substituted the term range by window size through the manuscript.

28) Results, line 219: what is meant by “manual curation of the data”? What does this curation involve? Are the authors simply manually editing the alignments (and if so then simply say this) or does this curation involve something else? It would help if further information was given here on how much this manual curation improved the estimates of RNA-editing. (From my reading of the later part of the Discussion I now presume that ‘manual curation’ simply means that the alignment is deemed to be good and that the variant is in dsRNA – if this is the case then this needs to be spelled out in the results. However, in what sense is this ‘manual curation’, since both the alignments and dsRNA prediction are done by a computer?).

We have added a segment in the methods to describe what the manual curation involves (121-126). In the case of the alignments, they are not edited, but revised. We run mafft “blindly” and in some cases, the alignment quality is not good enough or the edited amino acid is in a low-quality aligned portion. The same happens with the secondary structure. Linearfold gives us the secondary structure prediction for each sequence, but we have to check whether the edited nucleotide is in dsRNA or not.

29) Discussion, line 232: change “it is Adarb1 the gene expressed” to “it is the Adarb1 gene that is expressed” (but also note the incorrect gene name here as well, as highlighted in point 1 above).

 Changed the sentence (263-265).

30) Discussion, line 235: the word ‘earthquakes’ is not appropriate here. Change to something like ‘innovations’ or ‘elaborations’.

Removed.

31) Discussion, line 252: change “suffers” to “experiences”.

Changed (287).

32) Discussion, line 260: it is not appropriate to talk of ‘the’ amphioxus genome when actually several different amphioxus species have been sequenced and the authors are referring to a different species to the one that they have analysed in this manuscript. Thus, line 260 should be changed to something like, “The first report of an amphioxus genome, already indicated [32] that amphioxus genomes are highly polymorphic”.

Changed as suggested (294)

33) Discussion, line 270: change “choose” to “chose”.

Changed (304).

34) Discussion, line 273: space between “amino acids”.

Aminoacidic and aminoacid changed to amino acid though all the manuscript. Regarding the filtering, the reviewer is correct. We filter the transcripts only once, and the second description has been removed.

35) Discussion, line 276: add an ‘s’ to “events” in “events (Table 1)”.

Added (310).

36) Conclusion, lines 293-4: change “which astringency that can be finely adjusted” to something like “in which the stringency can be adjusted”.

Changed (333).

37) Queries about the pipeline…

In order to remove SNPs from the candidate SNV events the authors use the polymorphism data available in the lanceolatum genome sequence. Is this polymorphism data sufficient to do a good job of removing a high proportion of lanceolatum SNPs? That is, a population-level estimate of SNPs will presumably be much more extensive than the number of SNPs obtained from the deliberately limited number of SNPs produced from sequencing a single diploid individual to obtain the lanceolatum draft genome sequence. Although the authors allude to this in their discussion, it could be more explicitly highlighted and commented on, in terms of how this approach could be improved upon in future.

The available SNP data is not enough to remove all the SNPs from the SNVs. Although the draft genome sequence of B. lanceolatum was obtained from several individuals, it is still not enough to account for all the naturally occurring SNPs.

38) What is the availability of pipeline code?

It is available on github at https://github.com/mihizawi/Ortho-RNAediting

39) One limitation of the complete pipeline is that it focuses only on editing events that are similar (and potentially homologous?) to events found in human genes. RNA editing that occurs in a particular species more generally will not be assessed. This is unfortunate from an evolutionary perspective, as there will obviously be much evolution occurring via genes that either do not have human homologues, or can be edited at different locations from the human homologue. This aspect of the work could also be briefly discussed.

The reviewer is correct. In future works we plan to sequence both mRNA and DNA from same individuals to gain real insights about the editing in amphioxus. This in combination with this pipeline would allow us to use amphioxus in comparisons with other invertebrates to try to fill the gaps that our current approach has because of the distance between the used species. This now is briefly discussed in the text (319-322).

Perhaps (for future work?) the authors could run the pipeline by searching for octopus homologues that are edited alongside the human-focused analyses done here to help increase confidence in RNA-editing events that might be conserved across octopus, amphioxus and humans, and to provide a complimentary dataset of octopus-amphioxus events versus the amphioxus-human events.

This is in fact one of our future prospects!
